# Effects of a sport-based positive youth development program on youth life skills and entrepreneurial mindsets

**Leapetswe Malete**[1]*, **Daniel McCole**[2], **Tshepang Tshube**[3], **Thuso Mphela**[4], **Cyprian Maro**[5], **Clement Adamba**[6], **Juliana Machuve**[7], **Reginald Ocansey**[8]

1 Department of Kinesiology, Michigan State University, East Lansing, Michigan, United States of America,
2 Department of Sustainability, Michigan State University, East Lansing, Michigan, United States of America,
3 Department of Sport Sciences, University of Botswana, Gaborone, Botswana, 4 Department of Management, University of Botswana, Gaborone, Botswana, 5 Department Physical Education and Sport Sciences, University of Dar es Salaam, Dar es Salaam, Tanzania, 6 Department of Education and Leadership Studies, University of Ghana, Legon, Accra, Ghana, 7 Department of Mechanical and Industrial Engineering, University of Dar es Salaam, Dar es Salaam, Tanzania, 8 Department Physical Education and Sport Studies, University of Ghana, Legon, Accra, Ghana

* maletele@msu.edu

**Data Availability Statement:** The data underlying the results presented in the study are available from https://doi.org/10.6084/m9.figshare.17125589.

## Abstract

Sport-based life skills interventions offer compelling pathways to understanding the role of physical activity and sport on youth psychosocial and other development outcomes. This is because of evidence that shows the benefits of sport programs to health and well-being of youth, and more lately other areas such as academic achievement and various life skills such as teamwork, leadership and goal setting. However, much of the research in this area of youth development is largely descriptive, with limited capacity to infer causal relationships and application across contexts. Therefore, this study examines the effects of a sport-based intervention program on life skills and entrepreneurial mindsets of youth from three African countries (n = 146, average age = 15.9 years, female = 48.6%). Half of the recruited participants were assigned to a three-week life skills intervention program and the remaining half to a sport-only control program. Both groups completed a demographic information questionnaire, Life Skills for Sport Scale and the General Enterprising Tendency v2 test. Two-way mixed ANOVAs showed significant post-intervention changes in life skills for both groups but changes in entrepreneurial mindsets for the intervention group only. This demonstrates the relevance of sport-based interventions to youth development outcomes in different contexts and the transformative potential of youth sport reported in previous studies. The findings have important implications for intentional and targeted delivery of programs to enhance specific youth development outcomes.

## Introduction

Sport is widely recognized as a tool to keep youth constructively engaged and is valued for its health benefits and capacity to teach life skills such as goal-setting, emotional control,

**Funding:** This research was done with funding from the Alliance for African Partnership (AAP) at Michigan State University. The funder had no role in study design, data collection and analysis, decision to publish, or preparation of the manuscript. None of the authors received a salary from AAP.

**Competing interests:** The authors have declared that no competing interests exist.

leadership, self-discipline and resilience [1,2]. The number and scale of local and global physical activity (PA) or sport-based positive youth development (PYD) programs, such as 4-H, Right to Play and The First Tee, demonstrate decades long interest in sport and youth psychosocial development [3–5]. The rapid growth of PA and sport-based PYD agencies, and millions of youths they attract, affirm the power and intrinsic appeal of sport to children and youth, which makes them excellent avenues for enhancing youth development.

Over the past two decades, a body of knowledge has emerged on life skills transfer from PA and sport to a wide range of youth development outcomes such as academic achievement, self-discipline and interpersonal relations. Although many of the skills shown to be developed through sport and PA are also skills required of entrepreneurs (e.g., goal-setting, resilience, leadership and self-discipline), there is limited research about how structured youth sport programs affect youth entrepreneurial mindsets, an attribute with significant implications for improved livelihoods, especially in areas with high unemployment, or where it is difficult to find a career path in mainstream professions. In sub-Saharan Africa, for example, where a very large percentage of the population is comprised of children and young adults (a phenomenon labeled as the "youth bulge"), there has been a focus on job creation and entrepreneurship to ensure there are adequate employment opportunities as Africa's youth come of age [6,7].

Given the potential relationship between life skills development through participation in sport and the development of an entrepreneurial mindset, as well as the dearth of research in this area, this study seeks to broaden the scope of the literature by demonstrating a causal relationship between a sport-based PYD and entrepreneurship intervention program and the development of life skills and entrepreneurial mindsets in youth across three regions of Africa. Developing life skills and entrepreneurial thinking among youth could have more direct applications to economic opportunities, improved livelihoods and building resilient communities [8–11]. It also hopes to address some of the identified gaps in the PYD and sport-for-development research [12]. These gaps include evidence of causal relationships, application of findings across contexts, and investigations involving youth from understudied populations. The study also offers a different approach to nurturing entrepreneurial mindset in youth to what has widely been reported in the sport management and entrepreneurship research [13,14].

Many African countries face persistent challenges of youth unemployment, labor underutilization and youth disaffection [15]. These challenges are rooted in inadequate human and economic development which are considered to be among the leading drivers of forced internal and global migration [16]. Despite remarkable economic growth across the continent, there are significant concerns about the number of youths who are not in formal education, employment, or training across the Africa continent. For instance, the global unemployment rate is estimated to be between 13–30% and African countries, tend to be overrepresented on the higher end of this rate [17]. Therefore, mitigation of these systemic challenges through programs that specifically build youth resilience is urgent. Developing life skills and entrepreneurial mindsets would be one such mitigation strategy. Using a sport-based PYD program to nurture life skills and entrepreneurship is compelling because sport shares many skills with entrepreneurship. Examples articulated in the literature are, self-discipline, teamwork, leadership, goalsetting, risk taking, emotional control and resilience [1,9,18]. Possession of these skills and translating them into novel economic pathways could be transformative, especially to generations of youth with limited alternatives.

## Conceptual framework

This research was conceptualized within the broader PYD framework and positive values of sport. PYD offers an asset or strength-based view of youth development, where youths are

considered active agents of their own development and possessing the resources needed to achieve resilience and withstand difficult circumstances [9,19–21]. PYD can be viewed as both instrumental and aspirational in that it has been widely used as a philosophical orientation of many youth development policies and programs, as well as a guide in the actual delivery of programs [18,22]. The significant overlap between Lerner and colleagues' [19,23] Five C's (competence, confidence, connection, character, caring and compassion) and sport-related life skills (e.g., teamwork, goal setting, social kills, problem solving, emotional skills, leadership, time management and communication) [1,20] offers a compelling case for combining aspects of the two in youth sport and youth development research. The goal of sport-based PYD research is really to examine ways in which sport and physical activity settings can be used to optimize youths' internal strengths and positive sport experiences [1,9]. Therefore, the sport-PYD framework used in this study was guided by these principles and specifically a desire to: 1) develop assets youth can use to thrive, 2) enhance greater sense of autonomy in youth and 3) help to reorient the pervasive view of African youth as helpless and having deficits that require reduction, to a view that with good guidance and strengthened capacities, the youth can become agents of their own development and the development of their communities. Findings from recent investigations demonstrate the significant potential of this orientation and the use of PA or sport-based PYD programs [11,22]. However, many of the existing programs and related research are yet to reach the core of youth who need these programs the most. Doing this especially in less-studied communities and regions of the world where youth are faced with many development challenges remains key. Furthermore, using sport-based PYD to understand how various life skills and competencies associated with sport can be optimized and transferred to other domains of youth development is still highly needed. Application to the development of entrepreneurial mindsets of youth is one such example. Fig 1 presents an illustration of the conceptual framework and study variables.

## Entrepreneurial mindsets

The concept of entrepreneurship used in this study is in line with Howard Stevenson's definition that *"entrepreneurship is the pursuit of opportunity beyond resources controlled"* [24]. Consistent with the key words, pursuit, opportunity and resource constraints, we defined entrepreneurial mindset as the ability to recognize, passionately pursue and exploit opportunities through innovative and creative problem solving and risk mitigation even in the context of significant resource limitations. This approach of channelling passion in a constructive way to achieve some goal is also at the core of the competitive spirit of the athlete. Sport is considered

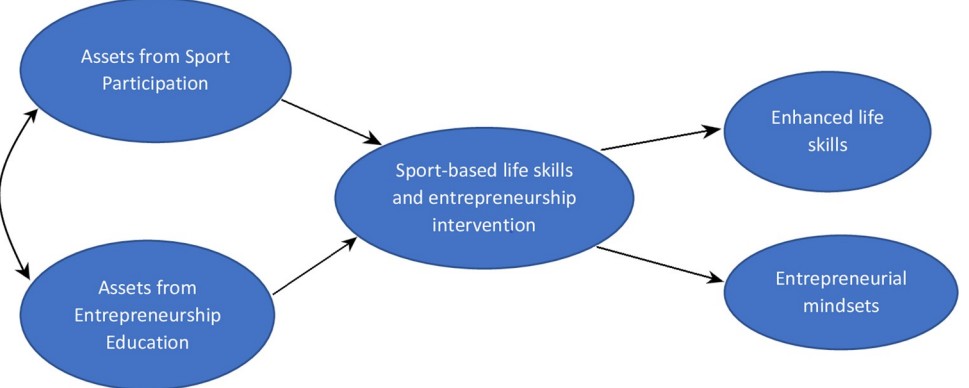

**Fig 1. Conceptual framework and study variables.**

to imbue various attributes associated entrepreneurial mindsets, such as creativity, innovation, risk taking, goal setting and resilience. Youth entrepreneurship is an emergent concept in global economics that is considered significant to economic development and social transformation [25]. Unfortunately, it is not widely taught in schools and in some cases, traditional education thwarts its development [25]. The Global Entrepreneurship and Development Institute [26] which considers key economic indicators like GDP to rank countries, ranked Botswana 52nd, Ghana 93rd and Tanzania 115th out of 137 countries in the global entrepreneurship index. The three countries rank much lower in the inclusion of entrepreneurship in their education systems. Research has shown that people from the lower end of the socio-economic spectrum (e.g., those earning low wages and with lower educational attainment) have improved their economic situations through entrepreneurship over similar groups who earn wages or salaries from employers [27]. Therefore, there are potentially significant gains to be derived from intentional development of life skills and entrepreneurial mindsets through sport-based PYD programs in these countries, especially considering their high youth unemployment and other development challenges.

Therefore, the purpose of this study was to examine if exposure to a sport-PYD and entrepreneurship intervention program will lead to improvements in life skills and entrepreneurial mindsets among youth from Botswana, Ghana and Tanzania. We also examined if these differences varied by gender and school level. The following hypotheses were tested:

1. Youth exposed to a sport-based PYD program will report significant pre to post-intervention improvements in various life skills compared to youth in a sport-only control program.

2. Youth exposed to a sport-based PYD program will report significant pre to post-intervention improvements in their entrepreneurial mindsets compared to youth in a sport-only control program.

The choice of samples of youth from three diverse regions of Africa: East Africa (Tanzania), West Africa (Ghana) and Southern Africa (Botswana) was intended to broaden the youth sport, PYD and entrepreneurship education research landscape. While this may not be apparent to an outsider, the three countries selected for this study have a shared colonial history, similar education systems, but represent distinct and very diverse sociocultural, economic and political contexts. Tanzania has a considerably larger population (59,734,218) compared to Ghana (31,072,940) and Botswana (2,351,627) [28]. These similarities and differences make the countries excellent choices for a multi-country/multi-region case study of effects of an intervention to enhance youth development outcomes if the goal is to determine potential application of the findings to other contexts.

## Materials and methods

### Participants

A total of 146 youth aged 12–20 years ($M = 15.88$, $SD = 1.66$) from a possible 150 recruited from 13 public middle schools (also called junior secondary) in Botswana, Ghana and Tanzania completed the study. About 51% of all the participants were male and they were all of African descent. Botswana had 43 participants (male = 23), aged 12–17 years ($M = 14.40$, $SD = 1.03$) from 5 schools in Gaborone, the capital. Twenty-five were assigned to the intervention group and 25 to the control group. All participants in the intervention group completed the study compared to 18 in the control group. In Ghana, 53 participants (male = 27) aged 13–20 years ($M = 16.91$, $SD = 1.57$) were recruited from 4 schools in Accra, the capital. Twenty-

**Table 1. Summary demographics by country.**

| County | Variable | | Intervention Group | Control Group |
|---|---|---|---|---|
| Botswana | Age [a] | | 14.36 (1.04) | 14.44 (1.04) |
| | Sex | Male | 11 (44.0%) | 12 (66.7%) |
| | | Female | 14 (56.0%) | 6 (33.3%) |
| | Botswana Sample | N | 25 | 18 |
| Ghana | Age [a] | | 16.81 (1.64) | 17.00 (1.52) |
| | Sex | Male | 14 (51.9%) | 13 (50.0%) |
| | | Female | 13(48.1%) | 13 (50.0%) |
| | Ghana Sample | N | 27 | 26 |
| Tanzania | Age [a] | | 16.21 (1.44) | 15.92 (0.95) |
| | Sex | Male | 12 (49.98%) | 13 (52.0%) |
| | | Female | 12 (49.98%) | 12(48.0%) |
| | Tanzania Sample | N | 25 | 25 |

[a] Mean (Standard Deviation) is reported for age.

seven were assigned to the intervention group and 26 to the control group. All participants completed the study. In Tanzania, 50 participants (male = 25) aged 14–19 years ($M = 16.06$, $SD = 1.22$) from 4 schools in Dar es Salaam, the capital, enrolled and all completed the study. Twenty-five were assigned to the intervention group and 25 to the control group. There we no significant differences between the intervention and control groups from the three countries on key demographic variables, namely: age, gender, type of sports played, history and frequency of sport participation. The top five sports played by the youth in each country were soccer (38.4%), basketball (14.4%), volleyball (9.6%), netball (7.5%) and track and field (6.8%). The youth in both groups reported playing sport on average three times a week and a had two to three years of experience playing their favorite sport. A summary of the sample's demographics by country is provided in Table 1.

### Study design and intervention

This study followed a quasi-experimental pre-posttest control group design. The intervention program was collaboratively developed by the principal investigators and country project teams. The research team had backgrounds in design and implementation of research and outreach programs in youth sport, life skills, pedagogy, entrepreneurship education and community sustainability. The intervention entailed sport-based life skills and entrepreneurship lessons and activities, simultaneously delivered over 22 days, divided into two overnight camps. The first camp was 14 days and the second one was eight days. The control program followed a similar camp structure, was run at the same time as the intervention but entailed typical sport activities. The groups camped in separate locations to minimize contact between participants and training staff. Costs related to the camp, including housing, meals and transportation from home to the venues and back were fully covered by the program. Both the intervention and control groups completed the survey tools on their first day of arrival at the camps, at the end of week two and on the last day of the second camp. The comparative data used in this study was from the first and last assessment because one country did not administer assessments at the end of week two. The camps were run during the school holidays.

A typical day program lasted seven hours from 9:00 am– 4:00 pm. In the evenings, the campers took part in a variety of guided social activities that included board games and movies. The sport and life skills component of the intervention program included games for social

and emotional development as well as practice and match sessions in soccer, volleyball and track and field. Examples of the social emotional games are, the human knot, captain, have you ever and the balance activity mirror [29]. The activities were selected to teach a variety of skills including teamwork, goal setting, interpersonal communication, problem-solving, emotional regulation and leadership. These skills were selected because of their relevance to youth development needs and also because they are clearly articulated in Life skills Scale for Sport (LSSS) which was selected for use in this study [30]. Age-appropriate experiential training workshops on topics such as market research, identifying entrepreneurial opportunities and financial literacy were used to teach entrepreneurial mindsets. Participants were guided to identify similarities between life skills from sport and skills identified as important to the success of entrepreneurs such as goal setting, calculated risk-taking, resilience and leadership. To enhance the learning of individual life skills and entrepreneurship, as well as demonstrate the connection between them, the daily program was structured to deliver these as separate and combined activities.

## Procedure

A target of 50 participants were recruited from public middle school athlete in the capitals of each of the three countries. Sample size estimation was done using Cohen (1992) statistical power analysis [31]. Assuming a small effect size (.2) from the intervention program at $\alpha = .05$ and a power of .80, we estimated that we will need at least 75 participants in the experimental and control group. Country project leaders accessed youth athlete rosters from secondary schools' athletic associations in each country from which they recruited participants from commonly played sports in the countries, namely track and field, soccer, basketball, and volleyball. Efforts were made to select an equal number of male and female athletes from the rosters of participating sports. The same approach was used to assign selected participants to the intervention and control group. Half of the selected participants were assigned to a sport-based life skills and entrepreneurship programs and the remaining half to a sport-only control program. Recruited participants were given these documents to take home to their parents: letters introducing the study, parent consent forms and youth assent forms. They were required to return signed parent consent forms and assent forms as a condition to participating in the study and doing overnight camps. Only youth who had been granted signed permission by their parents took part in the study.

## Ethical clearance

Permission to conduct this study was obtained from the Institutional Review Boards (IRB) of partner universities and the government ministries (departments) responsible for youth, sport and education in each country. Additional approvals were obtained from the schools. Consent forms contained a statement that made parents and participants aware of their rights to decline participation, withdraw from the study at any point or decline to answer any questions. All selected participants were made aware of these rights and that they would not be allowed to take part in the study and the camps without signed parental consent and against their will. All participants provided signed parent/guardian consent prior to completing the questionnaires and joining the camps.

## Measures

**Demographic information.** A demographic information questionnaire was used to collect data on age, gender, school grade and sport experiences.

**Life skills development.** Life skills were assessed using the LSSS [30]. The LSSS is a validated 43-item Likert-type scale ranging from 1 (not at all) to 5 (very much). Participants are asked to rate how much they think playing sport has helped them to develop or learn the various life skills represented by the different items. The stem for each item is, *"Please rate how much your sport has taught you to perform the skills listed below"* followed by statements that best represents each of the eight life skills: teamwork (7 items), goal-setting (7 items), social skills (5 items), problem-solving (4 items), emotional skills (4 items), leadership (8 items), time management (4 items) and communication (4 items). Cronbach's α for each of the eight subscales from the initial study of instrument's development and validation were above .70 [30]. Reliabilities from the current study were excellent at both pretest (.92) and posttest (.94) in the entire sample as well as in each sample by testing occasions and countries, ranging from .88 to .93. Therefore, the LSSS was considered a reliable measure of life skills for this study.

**Entrepreneurial mindsets.** The entrepreneurial mindsets were measured using the General Enterprising Tendency version 2 test (GET2). The initial instrument (GET) was developed in 1989 by Caird & Johnson using key psychological characteristics of entrepreneurs that were reported in the literature (e.g., need for achievement, need for autonomy, creative tendency, risk calculation, and internal locus of control) [32,33]. It was later refined to the General Measure of Enterprising Tendency version 2 (GET2) [34]. Use of the GET and GET2 can be widely found in the entrepreneurship literature [35–42], including studies conducted with people located in Global South countries [43–47], which is the context of the current study. The GET2 asks participants to indicate either "tend to agree" or "tend to disagree" with statements that correspond to each of the attributes. Points are added to determine the overall score which ranges between 0 and 54. Scores ranging between 44 and 54 represent very high enterprising qualities, 27 to 43 represent medium enterprising qualities and 0 to 26 represent low enterprising qualities. The reliabilities for the total GET2 scale were high at both pretest (.83) and posttest (.86) in the entire sample.

## Data analyses

Preliminary analyses of variance (ANOVAs) were run to test for gender and school level differences on life skills entrepreneurial mindset across groups. This was followed by hypothesis testing using 2 (group: intervention and control) × 2 (time: pre, post) mixed ANOVAs. "Time" was considered the within-subjects factor and "group" was the between-subjects factor. When the interaction of time and group was found to be significant, we tested the simple main effect of time on each group. Partial eta squared ($\eta_p^2$) were calculated as measures of effect size. Follow-up paired samples t-tests were examined in the case of significant effects. Bonferroni correction was used for multiple comparisons. Less than 1% of data was missing and missing values were considered to be missing completely at random and to have negligible influence on the findings. Therefore, pairwise deletion was allowed to handle all missing values during data analyses. All analyses were done using Statistical Package for Social Sciences (SPSS) (Version 24).

## Results

### Intervention effects

**Preliminary analyses.** To address the potential effects of various demographic variables on the study's outcome variables and decide whether to include them in hypothesis testing, we tested for age, gender, major sports played, history and frequency of sport participation and school level differences between the experimental and control group. We also tested for age,

gender and school level differences in life skills and entrepreneurial mindsets at pretest and posttest within and between countries. Results from these preliminary analyses showed no significant differences (p >.05). Therefore, there was no need to control for the effects on any of the outcome variables in hypotheses testing.

**Life skills.** The first hypothesis was that youth exposed to a sport-based PYD program will report significant pre to post-intervention improvements in various life skills compared to youth in a sport-only control program. To get a general overview of changes in life skills from pretest to posttest for the full sample, we ran a two-way mixed ANOVA with the total score of the LSSS. While the goal of this study was to examine changes in the eight subscale scores of the LSSS, the developers of the scale made a case for the viability of using the total score [30]. Results of the total score are summarized in Fig 2 and reveal that the group × time interaction was not significant for the total LSSS scores ($p$ = .36). We therefore proceeded to examine the main effects. The results showed significant main effects for time with the higher values of life skills at posttest, $F$ (1, 142) = 105.46, $p$ = .001, $\eta_p^2$ = .43), while the main effect of group was not significant ($p$ = .71). An examination of these intervention effects by country showed that only

### (a) Total LSSS Scores

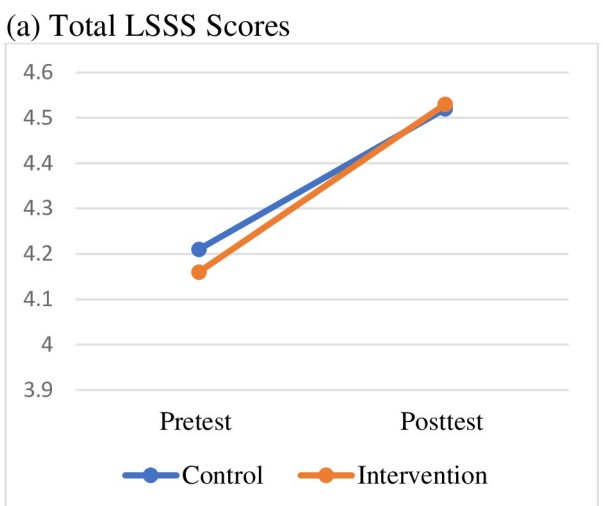

### (b) Botswana LSSS Scores

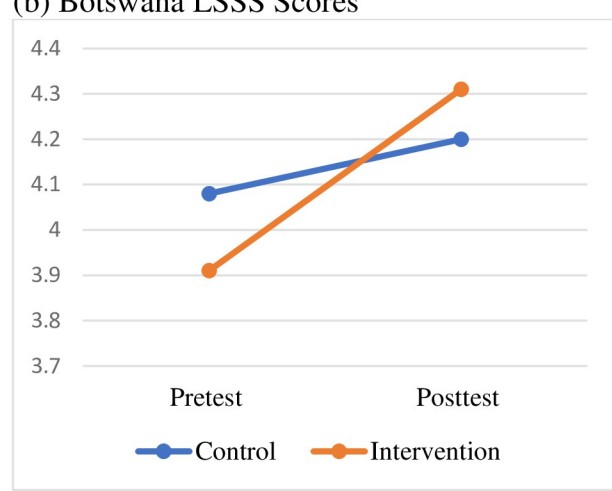

### (c) Ghana LSSS Scores

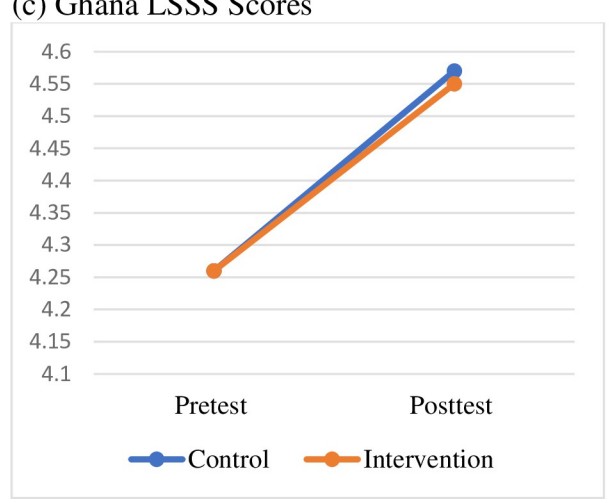

### (d) Tanzania LSSS Scores

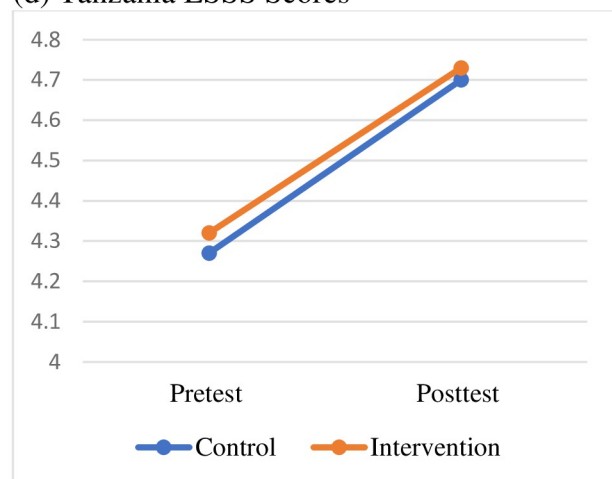

**Fig 2. Group and time differences for life-skills.**

Botswana had significant group × time interaction on the total score for life skills for sport (Pillai's Trace = .11, $F$ (1, 41) = 5.37, $p$ = .03, $\eta_p^2$ = .12), indicating a significant difference between the intervention and control groups in total life skills at pretest and posttest. Follow-up analysis of these effects using paired samples $t$-test showed that the pretest score for the control group ($M$ = 4.08, $SD$ = .54) was significantly higher than that of the intervention group ($M$ = 3.91, $SD$ = .50). However, the intervention group ($M$ = 4.31, $SD$ = .48) had significant changes in their life skills from pre to post-intervention ($t$ (24) = 4.40, $p$ = .001), whereas the change for the control group ($M$ = 4.20, $SD$ = .43) was not significant. These results indicate that, while the control group in Botswana had significantly higher life skills scores at pretest compared to the intervention group, these differences were altered after the intervention where the intervention group experienced a significantly higher change in their mean score compared to the control group.

An examination of the main effects for the Ghanaian and Tanzanian samples on the total life skills scores revealed significant main effects of time for the Ghanaian sample (Pillai's Trace = .47, $F$ (1, 51) = 45.58, $p$ = .001, $\eta_p^2$ = .47) and for the Tanzanian sample (Pillai's Trace = .50, $F$ (1, 46) = 47.33, $p$ = .001, $\eta_p^2$ = .51). Paired samples t-test for the two countries showed that both the intervention and control groups had significant increases in their life skills from pretest to posttest. This was unexpected and deserved closer examination to determine where the specific changes occurred at the level of each life-skill, which, in any case was the main objective of this study.

To test for hypothesized intervention effects on the eight life skills, two-way mixed ANOVAs were conducted on the pre and protests scores of the eight subscales for the total sample and at country level. In the total sample, the group × time interaction term was not significant on all LSSS subscales (.07 < $ps$ < 70). This suggests that the change in youth's subscales did not significantly differ by intervention and control groups. In the absence of significant group x time interaction effects across groups for the LSSS subscales, we proceed to examine the main effects for these factors for the total sample and at country level. This was important to determine subscale level differences as a consequence of the intervention. These results are summarized in Table 2.

An examination of change in the eight life skills subscale over time showed significant group by time interaction effects for emotional skills ($F$ (1, 41) = 14.02, $p$ = .001, $\eta_p^2$ = .26) and time management ($F$ (1, 41) = 11.05, $p$ = .002, $\eta_p^2$ = .21) in the Botswana sample. A follow-up examination of these interaction effects in emotional skills for the Botswana sample using paired samples $t$-test showed a pre to posttest improvement in emotional skills for the intervention group ($t$ (24) = 4.95, $p$ = .001), but not for the control group ($p$ = .81). Similarly, pre to posttest differences were observed for time management in the intervention group ($t$ (24) = 3.83, $p$ = .001) but not the control group ($p$ = .28). The Botswana control group had significantly higher means than the intervention group at pretest on both of these skills, but the intervention group had significant gains that even surpassed the control group at posttest. As summarized in Table 2, the results showed significant pre to post-intervention changes in the life skills for both the intervention and control groups in each country except for Teamwork ($p$ = .08) and Social Skills ($p$ = .07) in Botswana. Mean differences and standard deviation for the eight life skills sub-scales are presented in Table 3.

**Entrepreneurship.** The second objective of this study was to examine the effects of a sport-based PYD and entrepreneurship intervention program on youth entrepreneurial mindsets. Results from the two-way mixed ANOVA suggest that youth from the intervention group significantly improved their entrepreneurial mindsets from pretest to posttest compared to those who were in the control group, Pillai's Trace = .03, $F$ (1, 143) = 4.64, $p$ = .03, $\eta_p^2$ = .03.

**Table 2. Pre-post intervention differences in life skills' for total sample and by country.**

| Subscale | Country | Main Effects for Time | | | |
|---|---|---|---|---|---|
| | | $F$ | $df1, df2$ | $p$ | $\eta_p^2$ |
| Teamwork | Total Sample | 48.97 | 1, 142 | .001 | .26 |
| | Botswana | 3.14 | 1, 41 | .08 | .07 |
| | Ghana | 23.05 | 1, 51 | .001 | .31 |
| | Tanzania | 31.24 | 1, 46 | .001 | .40 |
| Goal Setting | Total Group | 36.74 | 1, 142 | .001 | .21 |
| | Botswana | 5.86 | 1, 41 | .02 | .13 |
| | Ghana | 8.59 | 1, 51 | .01 | .14 |
| | Tanzania | 32.62 | 1, 46 | .001 | .42 |
| Social Skills | Total Sample | 43.74 | 1, 142 | .001 | .24 |
| | Botswana | 3.54 | 1, 41 | .07 | .08 |
| | Ghana | 17.45 | 1, 51 | .001 | .26 |
| | Tanzania | 29.73 | 1, 46 | .001 | .39 |
| Problem Solving | Total Group | 68.45 | 1, 141 | .001 | .33 |
| | Botswana | 11.93 | 1, 41 | .001 | .23 |
| | Ghana | 15.91 | 1, 50 | .001 | .24 |
| | Tanzania | 47.21 | 1, 46 | .001 | .51 |
| Emotional Skills | Total Sample | 31.60 | 1, 141 | .001 | .18 |
| | Botswana* | 11.36 | 1, 41 | .001 | .22 |
| | Ghana | 13.01 | 1, 50 | .001 | .21 |
| | Tanzania | 7.35 | 1, 46 | .01 | .14 |
| Leadership | Total Sample | 37.19 | 1, 141 | .001 | .21 |
| | Botswana | 5.63 | 1, 41 | .02 | .12 |
| | Ghana | 12.40 | 1, 50 | .001 | .20 |
| | Tanzania | 21.63 | 1, 46 | .001 | .32 |
| Time Management | Total Sample | 44.92 | 1, 141 | .001 | .24 |
| | Botswana* | 8.51 | 1, 41 | .01 | .17 |
| | Ghana | 18.90 | 1, 50 | .001 | .27 |
| | Tanzania | 21.74 | 1, 46 | .001 | .32 |
| Communication | Total Sample | 26.48 | 1, 140 | .001 | .16 |
| | Botswana | 3.44 | 1, 41 | .07 | .08 |
| | Ghana | 11.57 | 1, 50 | .001 | .19 |
| | Tanzania | 16.59 | 1, 45 | .001 | .27 |

*Interaction effects for this factor reported previously.

Paired samples t-tests were conducted as a follow-up analysis of the significant effect. The result indicated that for the overall sample, youth who had gone through the intervention program showed significant improvement in their entrepreneurial mindsets ($t$ (75) = 3.10, $p$ = .001), whereas those who were in the control group did not ($p$ = .72). Mean differences are summarized in Table 4.

At the country level, only the Tanzanian sample had significant group × time interaction on entrepreneurship mindsets, Pillai's Trace = .22, $F$ (1, 47) = 13.20, $p < .001$, $\eta_p^2$ = .22. As indicated by a follow-up analysis using paired samples $t$-tests, the simple main effect of time was significant for the intervention group ($t$ (23) = 6.61, $p$ = .01) but not for the control group ($p$ = .19), implying the significant effect of intervention on entrepreneurial mindsets in the

**Table 3. Means for life-skills sub-scales by time and group across countries.**

| Variable | Country | Time | Group | M | SD |
|---|---|---|---|---|---|
| Teamwork | Total Sample | Pretest | Intervention | 4.24 | 0.54 |
| | | | Control | 4.60 | 0.38 |
| | | Posttest | Intervention | 4.29 | 0.51 |
| | | | Control | 4.54 | 0.37 |
| | Botswana | Pretest | Intervention | 4.31 | 0.45 |
| | | | Control | 4.23 | 0.64 |
| | | Posttest | Intervention | 4.40 | 0.45 |
| | | | Control | 4.44 | 0.40 |
| | Ghana | Pretest | Intervention | 4.44 | 0.79 |
| | | | Control | 4.02 | 1.53 |
| | | Posttest | Intervention | 4.53 | 0.33 |
| | | | Control | 4.57 | 0.45 |
| | Tanzania | Pretest | Intervention | 4.34 | 0.62 |
| | | | Control | 4.25 | 0.52 |
| | | Posttest | Intervention | 4.66 | 0.33 |
| | | | Control | 4.74 | 0.23 |
| Goal setting | Total Sample | Pretest | Intervention | 4.17 | 0.56 |
| | | | Control | 4.28 | 0.56 |
| | | Posttest | Intervention | 4.48 | 0.45 |
| | | | Control | 4.51 | 0.42 |
| | Botswana | Pretest | Intervention | 3.89 | 0.70 |
| | | | Control | 3.93 | 0.87 |
| | | Posttest | Intervention | 4.26 | 0.59 |
| | | | Control | 4.16 | 0.55 |
| | Ghana | Pretest | Intervention | 4.50 | 0.40 |
| | | | Control | 4.37 | 0.57 |
| | | Posttest | Intervention | 4.51 | 0.35 |
| | | | Control | 4.52 | 0.41 |
| | Tanzania | Pretest | Intervention | 4.27 | 0.45 |
| | | | Control | 4.38 | 0.34 |
| | | Posttest | Intervention | 4.66 | 0.27 |
| | | | Control | 4.68 | 0.23 |
| Social Skills | Total Sample | Pretest | Intervention | 4.01 | 0.77 |
| | | | Control | 3.97 | 0.82 |
| | | Posttest | Intervention | 4.39 | 0.64 |
| | | | Control | 4.41 | 0.54 |
| | Botswana | Pretest | Intervention | 3.80 | 0.89 |
| | | | Control | 3.93 | 0.87 |
| | | Posttest | Intervention | 4.05 | 0.82 |
| | | | Control | 4.16 | 0.55 |
| | Ghana | Pretest | Intervention | 4.39 | 0.50 |
| | | | Control | 4.33 | 0.42 |
| | | Posttest | Intervention | 4.45 | 0.44 |
| | | | Control | 4.46 | 0.46 |
| | Tanzania | Pretest | Intervention | 4.26 | 0.57 |
| | | | Control | 3.85 | 0.90 |
| | | Posttest | Intervention | 4.70 | 0.39 |

(*Continued*)

**Table 3.** (*Continued*)

| Variable | Country | Time | Group | *M* | *SD* |
|---|---|---|---|---|---|
| | | | Control | 4.55 | 0.55 |
| Problem Solving | Total Sample | Pretest | Intervention | 3.95 | 0.69 |
| | | | Control | 4.09 | 0.71 |
| | | Posttest | Intervention | 4.54 | 0.51 |
| | | | Control | 4.48 | 0.51 |
| | Botswana | Pretest | Intervention | 3.57 | 0.81 |
| | | | Control | 4.15 | 0.62 |
| | | Posttest | Intervention | 4.26 | 0.61 |
| | | | Control | 4.13 | 0.56 |
| | Ghana | Pretest | Intervention | 4.53 | 0.48 |
| | | | Control | 4.47 | 0.51 |
| | | Posttest | Intervention | 4.63 | 0.44 |
| | | | Control | 4.50 | 0.51 |
| | Tanzania | Pre | Intervention | 4.08 | 0.52 |
| | | | Control | 3.96 | 0.81 |
| | | Post | Intervention | 4.71 | 0.34 |
| | | | Control | 4.72 | 0.29 |
| Emotional Skills | Total Sample | Pretest | Intervention | 4.04 | 0.70 |
| | | | Control | 4.19 | 0.71 |
| | | Posttest | Intervention | 4.37 | 0.62 |
| | | | Control | 4.46 | 0.58 |
| | Botswana | Pretest | Intervention | 3.72 | 0.81 |
| | | | Control | 3.94 | 1.01 |
| | | Posttest | Intervention | 4.26 | 0.61 |
| | | | Control | 4.17 | 0.67 |
| | Ghana | Pretest | Intervention | 4.30 | 0.54 |
| | | | Control | 4.46 | 0.50 |
| | | Posttest | Intervention | 4.33 | 0.58 |
| | | | Control | 4.42 | 0.50 |
| | Tanzania | Pretest | Intervention | 4.32 | 0.52 |
| | | | Control | 4.54 | 0.44 |
| | | Posttest | Intervention | 4.65 | 0.32 |
| | | | Control | 4.70 | 0.52 |
| Leadership | Total Sample | Pretest | Intervention | 4.21 | 0.61 |
| | | | Control | 4.26 | 0.51 |
| | | Posttest | Intervention | 4.58 | 0.44 |
| | | | Control | 4.46 | 0.56 |
| | Botswana | Pretest | Intervention | 3.86 | 0.72 |
| | | | Control | 4.09 | 0.56 |
| | | Posttest | Intervention | 4.40 | 0.59 |
| | | | Control | 4.00 | 0.75 |
| | Ghana | Pretest | Intervention | 4.62 | 0.43 |
| | | | Control | 4.63 | 0.35 |
| | | Posttest | Intervention | 4.61 | 0.34 |
| | | | Control | 4.65 | 0.34 |
| | Tanzania | Pretest | Intervention | 4.35 | 0.39 |
| | | | Control | 4.30 | 0.42 |

(*Continued*)

**Table 3.** (Continued)

| Variable | Country | Time | Group | *M* | *SD* |
|---|---|---|---|---|---|
| | | Posttest | Intervention | 4.73 | 0.25 |
| | | | Control | 4.62 | 0.40 |
| Time Management | Total Sample | Pretest | Intervention | 4.21 | 0.74 |
| | | | Control | 4.23 | 0.69 |
| | | Posttest | Intervention | 4.66 | 0.55 |
| | | | Control | 4.63 | 0.47 |
| | Botswana | Pretest | Intervention | 3.76 | 0.92 |
| | | | Control | 3.97 | 0.72 |
| | | Posttest | Intervention | 4.43 | 0.80 |
| | | | Control | 4.17 | 0.58 |
| | Ghana | Pretest | Intervention | 4.77 | 0.27 |
| | | | Control | 4.75 | 0.29 |
| | | Posttest | Intervention | 4.71 | 0.34 |
| | | | Control | 4.76 | 0.25 |
| | Tanzania | Pretest | Intervention | 4.38 | 0.51 |
| | | | Control | 4.30 | 0.79 |
| | | Posttest | Intervention | 4.83 | 0.29 |
| | | | Control | 4.84 | 0.30 |
| Communication | Total Sample | Pretest | Intervention | 4.32 | 0.72 |
| | | | Control | 4.39 | 0.75 |
| | | Posttest | Intervention | 4.66 | 0.54 |
| | | | Control | 4.62 | 0.49 |
| | Botswana | Pretest | Intervention | 4.17 | 0.77 |
| | | | Control | 4.04 | 0.97 |
| | | Posttest | Intervention | 4.47 | 0.76 |
| | | | Control | 4.22 | 0.61 |
| | Ghana | Pretest | Intervention | 4.70 | 0.37 |
| | | | Control | 4.60 | 0.42 |
| | | Posttest | Intervention | 4.68 | 0.40 |
| | | | Control | 4.63 | 0.41 |
| | Tanzania | Pretest | Intervention | 4.57 | 0.39 |
| | | | Control | 4.54 | 0.59 |
| | | Posttest | Intervention | 4.83 | 0.27 |
| | | | Control | 4.90 | 0.18 |

Tanzanian sample. Strangely, the Ghanaian sample had relatively low pretest and posttest entrepreneurship scores for the two groups compared to Botswana and Tanzania samples, suggesting possible contextual differences. The country differences are illustrated in Fig 3.

## Discussion

The objectives of this study were to examine if exposure to a sport-based PYD intervention program will lead to significant improvements in life skills and entrepreneurial mindsets in youth. In the process we also examined if data from three regions of Africa would support the extant literature on life skills and PYD, especially the potential effects of sport-based life skills

**Table 4. Descriptive statistics of entrepreneurship by time and group across countries.**

| Country | Time | Group | n | M | SD |
|---|---|---|---|---|---|
| Total Sample | Pretest | Intervention | 76 | 26.75 | 7.71 |
| | | Control | 69 | 26.65 | 9.33 |
| | Posttest | Intervention | 76 | 28.49 | 8.65 |
| | | Control | 69 | 26.38 | 8.59 |
| Botswana | Pretest | Intervention | 25 | 32.80 | 5.20 |
| | | Control | 18 | 33.44 | 6.85 |
| | Posttest | Intervention | 25 | 32.72 | 5.22 |
| | | Control | 18 | 29.33 | 4.10 |
| Ghana | Pretest | Intervention | 27 | 18.67 | 4.55 |
| | | Control | 26 | 16.31 | 4.15 |
| | Posttest | Intervention | 27 | 18.70 | 4.85 |
| | | Control | 26 | 17.31 | 4.38 |
| Tanzania | Pretest | Intervention | 24 | 29.54 | 4.19 |
| | | Control | 25 | 32.52 | 3.02 |
| | Posttest | Intervention | 24 | 35.08 | 3.46 |
| | | Control | 25 | 33.68 | 5.09 |

training on other domains of life [7–9]. We considered the use of a sport-PYD intervention program to develop entrepreneurial mindsets in youth in three different countries as not only novel, but also significant to demonstrating the potential application of the study's approach and its findings in different youth populations representing East Africa, West Africa and Southern Africa.

Data from this study partially supported the hypothesized causal relationship between the sport-based intervention and changes in life skills and entrepreneurial mindsets. One of the primary objectives of this study was to determine changes in life skills as a consequence of a sport-based intervention program. Our study showed no significant interaction effects on the total life skills score. Although the control group had a slightly higher total mean score than the intervention group at pretest, both groups had significant increases in their life skill scores at pretest. The absence of significant interaction effects and presence of main effects for time in the total LSSS subscale scores for both the intervention and control groups are consistent with previous studies on life skills through sport which reported that implicit and explicit teaching and learning of life skills occur even when unplanned in youth sport programs [5,8,23]. The literature suggests that implicit or indirect teaching and learning of life skills happens when coaches are adept at making teaching, learning and modeling of life skills part of mainstream activities [5,23]. The findings suggest that implicit and explicit learning happened in this study.

In this study, the control and intervention groups in the three participating countries went through simultaneous camps of structured day and evening activities. The only difference between the two groups was that the intervention group received life skills and entrepreneurship education through sport while the control group did the usual adult supervised and structured sport activities. Two possible explanations could be used to account for these effects. First, broad awareness of the overarching objectives of our study among control group coaches could have led them to model PYD behaviors that may have optimized youth life skills, even though the control group camp locations were separate from the intervention group camps. Second, and in line with previous findings, it is conceivable that, when well organized and structured with the development of youth in mind, youth sport camps may be as beneficial to

(a) Total Entrepreneurship Scores

(b) Botswana Entrepreneurship Scores

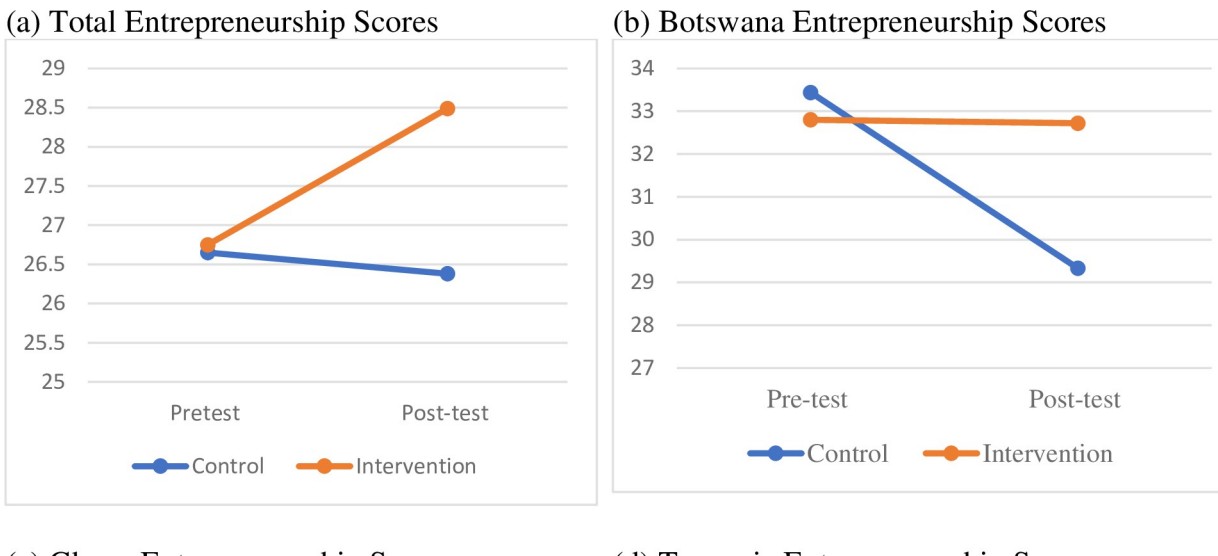

(c) Ghana Entrepreneurship Scores

(d) Tanzania Entrepreneurship Scores

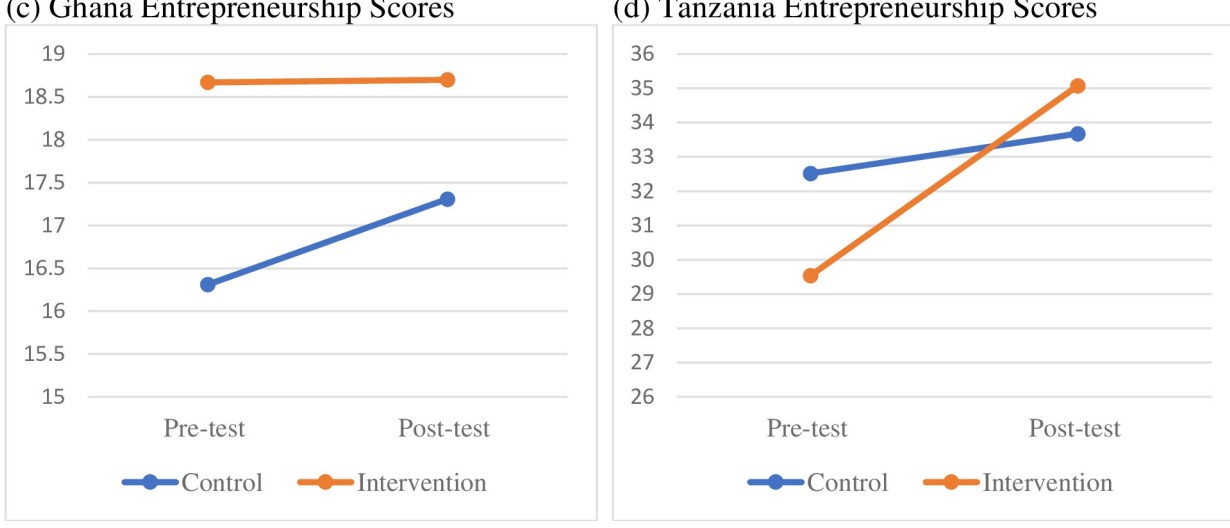

**Fig 3. Group and time differences for entrepreneurship.**

life skills development as programs that are designed specifically to develop life skills. Indirect teaching and learning of life skills may be inherent to sport camps because of greater bonding among youth participants and staff and less emphasis on competitive ethos that pervade competitive and elite sport. The significant changes in the intervention group suggest that specific training on life skills to coaches and program staff who deliver programs may augment these skills.

This study's use of a pre-post control group design to examine the effects of a sport-based PYD program on life skills and entrepreneurial mindsets among youth drawn from three countries is novel. Therefore, it was compelling to examine which of the target variables changed the most, as well as country level differences in these changes. An examination of the LSSS factors showed that Leadership, Goal Setting, Problem Solving and Time Management had significant post-intervention changes. This is consistent with what has been reported in previous studies about the direct and indirect learning of these skills from sport [9,30,48,49]. Changes in Emotional Skills in the Ghanaian sample were significant for the control group but

not for the intervention group. Although the mean for Emotional Skills did not change significantly in the Ghana intervention group, it was slightly higher at posttest, an indicator that the intervention was working. Botswana's control group had significantly higher pretest scores on Emotional Skills and Time Management compared to the intervention group, but there was a significant increase in these factors for the intervention group at posttest to an extent that the scores surpassed the control group scores. Curiously, there were no significant pre to posttest changes in Teamwork, Social Skills and Communication for the Botswana sample even though the scores trended positively for both groups. However, these skills changed significantly from pre to posttest in the Ghanaian and Tanzanian samples. These variations are to be expected and could be explained by differences in instructional style and how well the program was received by the youths in different contexts.

Along with Leadership, Teamwork, Social Skills and Communication are considered core aspects of social interactions in sports contexts. They were reported as the skills most learnt by the participants by Cronin and Allen [30]. Findings from the current study support this conclusion. The findings also suggest variations exist in how well the skills are learnt or prioritized in different contexts. Not to be ruled out are the context and timing of measurement of some these skills. For instance, it is likely that Emotional Skills may be susceptible to time, day and incident prior to testing than other life skills. This might have influenced group and country level differences. Other skills such as Leadership, Teamwork, Communication and Social Skills are likely to be more stable over time and less susceptible to minor events and changes.

This study also sought to examine if the sport-based PYD intervention had an effect on entrepreneurial mindsets of youth participants. Our results show that overall, exposing youth to a sport-based entrepreneurship intervention program leads to significant improvement in entrepreneurial mindsets. However, country level comparisons showed that the Tanzanian sample had the most effects, the Ghanaian sample had marginal improvements and scores from Botswana's sample declined at posttest. While caution has to be exercised in the interpretation of these findings because of country-level variations and a small sample, the findings offer support to what has been reported in previous studies on entrepreneurship education in sports context [50,51]. Although Gonzalez-Serrano and colleagues' [52] study was a cross-sectional survey of college students majoring in sport science, their findings offer insights on the relevance of intentionally nurturing entrepreneurial mindsets or making them part of the curriculum and educational policies. They found higher entrepreneurial self-efficacy, which they defined as higher perceptions about entrepreneurial abilities, to be a significant predictor of entrepreneurial mindsets. In line with this finding, we believe that exposing youth in our study to a sport-PYD and entrepreneurship intervention led to an overall increase in their self-perceptions of entrepreneurial capabilities, especially in Ghana and Tanzania. This is significant especially considering that entrepreneurial education is not part of the public school and college curriculum in the participating countries.

While caution has to be exercised about the interpretation of country level differences because of a small sample, the fact that this study showed significant differences in changes on entrepreneurial mindsets is worth mentioning because the differences suggest potential contextual factors may underlie these effects. The control groups in Botswana and Tanzania started off slightly higher on entrepreneurial mindsets compared to the intervention groups, but the intervention group significantly overshot the control group at posttest in Tanzania, while the Botswana sample saw no improvements. In Ghana, the intervention group was higher than the control group at pretest and both groups had marginal improvements over time. Interestingly, the two Ghana groups rated themselves relatively lower at pretest and posttest compared to the other countries. These differences could be due to any of the many complex factors that underlie entrepreneurial mindsets. Stated examples are a country's GDP,

entrepreneurship education, policies, social norms, infrastructure, the informal economy, access to finance and markets [27,52,53]. Our data suggests that overall, structured intervention programs are beneficial to the development of entrepreneurial mindsets, even when the likelihood of country variations in entrepreneurship is high.

The findings of this study also offer support for the transformative potential of sport-for-development programs reported in research done in other countries [11,54,55]. For instance, in their study of experiences of youth in a South African sport-for-development soccer program, Draper and Coalter [55] reported that participants had improved self-beliefs, empowerment and security after attending the program. These and other attributes are worthy of further examination alongside entrepreneurial mindsets within the context of life skills transfer to other youth development outcomes.

## Limitations

This study has some limitations that are worth noting. First, we were not able to assess youth's participation in previous after school programs, the content of such programs and how that could have affected our findings. While the kinds of intervention programs like the one we used in this study are relatively few and highly inaccessible to most of the youth in our study, the potential effects of various kinds of afterschool programs on the findings cannot be ruled out. Although we could not control for all possible extraneous variables, the use of camps and pre-posttest design make us confident about our findings. Second, the study could have benefited from a third comparison group made of youth who were neither in the intervention or control program. This would have allowed us to observe the magnitude of the intervention effects, implicit learning and sport camps on life-skills. Finally, our relatively small sample size, although not unusual in experimental designs like the one used in this study, could have limited the precision of our findings.

## Conclusions

Overall, our findings offer a compelling roadmap for ways in which well-structured sport based programs could be used to nurture entrepreneurial mindsets, life skills and other youth development outcomes reported in previous studies [9,10]. This line of work is particularly important given the limited research on causal relationship between assets from sport and entrepreneurial mindsets. Based on the pre to posttest changes in life skills among the intervention and control group, we conclude that youth are likely to acquire life skills from participation in sport even if the teaching of such skills is not intentional or the goal of sport activities. We also conclude, based on this study's results, that the development of more specialized skill sets such as entrepreneurial mindsets from sport is likely to be enhanced in a context where the teaching of such skills is intentional. The findings of our study suggest that exposing youth to after school enrichment programs and targeting the development of entrepreneurial mindsets as part of regular physical activity and sport could address some of the gaps in youth education and development that are rarely addressed through the regular school curriculum.

Future studies should consider expanding this line of research but with larger samples in participating countries, exploring more robust experimental designs, using mixed methods approaches and following the youth over a longer period of time. A mixed methods approach will be ideal to capture participants' perspectives, feelings, thoughts and emotions about the specific skills they learned the most and why. Using a comparison group of youth who are not doing any camps or non-sport participants could strengthen the design and underscore group differences. Longitudinal and follow-up studies will help with determining knowledge

retention and transfer to other tangible skills sets such as academic achievement and building startups. Researchers could also collaborate with development partners running similar youth programs, such as the United Nations Development Program and the World Bank to evaluate their intervention programs and strengthen the strategic alignment to youth and community livelihoods and Sustainable Development Goals.

## Supporting information

**S1 File.**
(DOCX)

## Author Contributions

**Conceptualization:** Leapetswe Malete, Daniel McCole, Tshepang Tshube, Thuso Mphela, Cyprian Maro, Clement Adamba, Juliana Machuve, Reginald Ocansey.

**Data curation:** Leapetswe Malete, Tshepang Tshube, Thuso Mphela, Cyprian Maro, Clement Adamba, Juliana Machuve, Reginald Ocansey.

**Formal analysis:** Leapetswe Malete.

**Funding acquisition:** Leapetswe Malete, Daniel McCole.

**Investigation:** Leapetswe Malete, Tshepang Tshube, Thuso Mphela, Cyprian Maro, Clement Adamba, Juliana Machuve, Reginald Ocansey.

**Methodology:** Leapetswe Malete, Daniel McCole, Tshepang Tshube, Thuso Mphela, Reginald Ocansey.

**Project administration:** Leapetswe Malete.

**Supervision:** Tshepang Tshube, Juliana Machuve, Reginald Ocansey.

**Writing – original draft:** Leapetswe Malete.

**Writing – review & editing:** Leapetswe Malete, Daniel McCole, Tshepang Tshube, Thuso Mphela, Cyprian Maro, Clement Adamba, Juliana Machuve, Reginald Ocansey.

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
