## [Decision Letter · Decision Letter 0]

5 Jul 2021

PONE-D-21-03687

Effects of a Sport-Based Positive Youth Development Program on Youth Life skills and Entrepreneurial Mindsets

PLOS ONE

Dear Dr. Malete,

Thank you for submitting your manuscript to PLOS ONE. After careful consideration, we feel that it has merit but does not fully meet PLOS ONE’s publication criteria as it currently stands. Therefore, we invite you to submit a revised version of the manuscript that addresses the points raised during the review process.

We look forward to receiving your revised manuscript.

Kind regards,

Ali B. Mahmoud, Ph.D.

Academic Editor

PLOS ONE

Journal Requirements:

2. Thank you for stating the following in the Financial Disclosure and in the Acknowledgments Section of your manuscript:

[Financial Disclosure:The authors would like to thank TheAlliance for African Partnership at Michigan State University for funding this project.

Acknowledgements: The authors would like to thank the University of Botswana, University of Ghana and University of Dar es Salaam for providing the resources needed to conduct the study.]

 [The funders had no role in study design, data collection and analysis, decision to publish, or preparation of the manuscript.]

Reviewers' comments:

Reviewer's Responses to Questions

**Comments to the Author**

1. Is the manuscript technically sound, and do the data support the conclusions?

Reviewer #1: No

Reviewer #2: Partly

2. Has the statistical analysis been performed appropriately and rigorously? 

Reviewer #1: No

Reviewer #2: Yes

3. Have the authors made all data underlying the findings in their manuscript fully available?

Reviewer #1: Yes

Reviewer #2: Yes

4. Is the manuscript presented in an intelligible fashion and written in standard English?

Reviewer #1: Yes

Reviewer #2: Yes

5. Review Comments to the Author

Reviewer #1: I enjoyed reading this manuscript reporting a study on a sports-based intervention with the aim to enhance life skills and entrepreneurial mindset. I think it addresses an interesting topic and it could present a valuable contribution to the literature. However, I also came across a number of (major and minor) points that should be addressed to enhance its contribution. I will address these roughly in order of their appearance in the manuscript.

(1) The introduction is quite long and might benefit from streamlining. In the beginning, the rationale for the study is not stated very clearly and I think this could also benefit from presenting the arguments in a more concise manner.

(2) There are no specific hypotheses presented, in particular not regarding the subscales of the life skills measure, which is in contrast to what is presented in the results section. If the main goal is, in fact, to study the effects on the subscales, this should be argued in the introduction and specific hypotheses should be put forward.

(3) The age range from 12 to 20 seems quite wide. What were the reasons for this wide age range and what are potential consequences?

(4) How were participants recruited?

(5) It seems that the numbers of participants (in particular those of the dropouts) across the subsamples do not add up to the total sample. Please check these numbers. It would also be helpful to describe how missing data was handled.

(6) On p. 8, it is described that intervention and control group were "very similar" on several variables. This should also be tested.

(7) There is conflicting information provided about how participants were assigned to the intervention or the control group (randomly vs. matched for gender and school level). How was this done exactly?

(8) Did I understand it correctly that all participants were active athletes? To what extend does this influence which conclusions can be drawn from the study?

(9) Did participants have to pay for camp participation or were there any incentives for participating in the study?

(10) Were there any considerations regarding statistical power or how was the sample size determined?

(11) In the measure section, it is stated that PYD was assessed. In my view, it would be more accurate to refer to life skills here as this was actually assessed.

(12) The low subscale reliabilities for the GET2 should nonetheless be reported. Are they in line with previous literature on the measure, or could they also indicate poor data quality?

(13) Exact p-values should be reported instead of "p < .05".

(14) Figure 1: What types of error bars are displayed here?

(15) In general, the results section needs strong improvement. As it is, it is very hard to follow and to determine which results are actually relevant for the research questions. I would suggest to streamline the results section, to focus on the time*group interactions, which are what is actually hypothesized (in my understanding) and also to refrain from reporting non-significant trends.

(16) There are a lot of tests reported. How are the effects of multiple testing accounted for?

(17) The discussion fails to address the finding that the total scores on life skills did not improve. This should be elaborated in more detail.

Reviewer #2: In general, this is an interesting and attractive subject. However, there are a few areas that need redress. Kindly improve on the connect/links between paragraphs and literature. Let it flow. Strengthen the global and local context from the three African countries. Bring out cases in the sport-based youth Development Programs that exhibited success/failure from both global and local perspectives. Try to inform your reader how PYD play an important role in supporting the all-inclusive development of youths.

The author(s) need to support the discussion about the objectives of the paper. The author's discussion, findings and conclusions should answer the key objectives of the study. The objectives appear to be thrown around. Kindly state the objectives clearly and address them in your findings and conclusion. Avoid mixing them.

In your discussion, Try to inform your reader on how widely you researched the topic on which you're writing. This is very important for the bibliographic studies

In addition, the authors may wish to link their study with the current need of international development agencies which have refocused their attention on strategies to enhance livelihoods (i.e, the youths in the three countries can improve), including the UN Development Program and World Bank. Indeed, for example the United Nations (2020) emphasizes livelihoods directly in relation to three of its Sustainable Development Goals (SDGs), e.g. including Goal 1: no poverty.

The authors may further refer to the manuscript to redress a few issues and typos

1. On line 58; Write PYD in full for the first time, then abbreviate

2. 107; The author may need have a diagram illustrating the Conceptual Framework in which the relevant study variables are clearly shown. Which data are you dealing with??? If you don’t have to show your conceptual framework, then use econometric models if possible.

3. 359; The authors may delete "test was"

4. The authors may check for more spelling error and spacing. there are many

This paper is good if the above comments are addressed.

6. PLOS authors have the option to publish the peer review history of their article (what does this mean?). If published, this will include your full peer review and any attached files.

Reviewer #1: No

Reviewer #2: **Yes: **Aziz Wakibi

---

## [Author Response · Author response to Decision Letter 0]

7 Oct 2021

We have attached a document that offers a point by point response to review comments.

---

## [Decision Letter · Decision Letter 1]

3 Dec 2021

PONE-D-21-03687R1Effects of a Sport-Based Positive Youth Development Program on Youth Life skills and Entrepreneurial MindsetsPLOS ONE

Dear Dr. Malete,

Thank you for submitting your manuscript to PLOS ONE. After careful consideration, we feel that it has merit but does not fully meet PLOS ONE’s publication criteria as it currently stands. Certainly, a few minor issues highlighted by Reviewer #1 will need to be resolved before acceptance. Therefore, I invite you to submit a revised version of the manuscript that addresses the points raised during the review process. In addition, whilst I am aware that you shared a link to the dataset in the original submission, your revision did not include that. So, please ensure your revised manuscript offers access to the study data.

We look forward to receiving your revised manuscript.

Kind regards,

Ali B. Mahmoud, Ph.D.

Academic Editor

PLOS ONE

Journal Requirements:

Reviewers' comments:

Reviewer's Responses to Questions

**Comments to the Author**

1. If the authors have adequately addressed your comments raised in a previous round of review and you feel that this manuscript is now acceptable for publication, you may indicate that here to bypass the “Comments to the Author” section, enter your conflict of interest statement in the “Confidential to Editor” section, and submit your "Accept" recommendation.

Reviewer #1: (No Response)

Reviewer #2: All comments have been addressed

2. Is the manuscript technically sound, and do the data support the conclusions?

Reviewer #1: Partly

Reviewer #2: Yes

3. Has the statistical analysis been performed appropriately and rigorously? 

Reviewer #1: Yes

Reviewer #2: Yes

4. Have the authors made all data underlying the findings in their manuscript fully available?

Reviewer #1: Yes

Reviewer #2: Yes

5. Is the manuscript presented in an intelligible fashion and written in standard English?

Reviewer #1: Yes

Reviewer #2: Yes

6. Review Comments to the Author

Reviewer #1: I want to congratulate the authors on a thorough revision of the manuscript. I think that the changes made helped strengthen the contribution of the manuscript. I only have a few minor suggestions to make:

(1) Data availability: If I didn’t miss anything, you state that the data set is available but not where it can be accessed.

(2) Power: Thank you for adding a power analysis. However, a lot of your discussion focuses on differences between the three samples – something that you did not plan for in your power analysis. I would suggest to substantially shorten the discussion of the differences between the three samples as the sample sizes in each country do not seem sufficient for drawing meaningful conclusions on this question.

(3) A minor thing but I thought it was confusing that the samples a referred to by the country name. I would suggested referring to the „Ghanaian sample“ or „sample collected in Ghana“ instead of just „Ghana“, for instance.

Reviewer #2: The current text of authors' manuscript suggests that all comments highlighted in the earlier review have been satisfactorily addressed. The manuscript is technically sound in its current state as data appear to support the conclusions. I am certainly convinced that this manuscript is now fit for publication by this journal.

7. PLOS authors have the option to publish the peer review history of their article (what does this mean?). If published, this will include your full peer review and any attached files.

Reviewer #1: No

Reviewer #2: No

---

## [Author Response · Author response to Decision Letter 1]

6 Dec 2021

Reviewer_1 Technical Comment #1

Data availability: If I didn’t miss anything, you state that the data set is available but not where it can be accessed.

Authors’ Response:

Apologies for not sharing the link. It is: https://doi.org/10.6084/m9.figshare.17049860

Reviewer_1 Technical Comment #2 

Power: Thank you for adding a power analysis. However, a lot of your discussion focuses on differences between the three samples – something that you did not plan for in your power analysis. I would suggest to substantially shorten the discussion of the differences between the three samples as the sample sizes in each country do not seem sufficient for drawing meaningful conclusions on this question

Authors’ Response:

Your concern about power analyses is appreciated. We have added text to the discussion on country comparisons, to explain how sample size limitations may limit generalizability of findings beyond the current sample. We noted that Power calculations are used as a tool to help researchers when designing studies and determining sample sizes needed to avoid making a Type 2 error (i.e., the intervention had an effect, but the sample size was too small to detect it). In our case the intervention had effects that we believe are worth reporting for the following reasons: 

1) This study used a fairly novel and exploratory approach with samples drawn from three countries to test the utility of a sport-base PYD intervention. Because of this there was no good way to determine the effect size needed for power calculations. This is why we used Cohen’s d, for power calculation because it is recommended when a literature review or pilot study cannot inform effect size (Cohen, 1988). 

2)The intervention described in this program was resource intensive as it involved 14-day and 8-day overnight camp experiences in three different countries. This meant the sample size of around 50 in each country was determined to be reasonable to detect the expected effects from the intervention.

We have provided additional rationale for including country differences in the discussion in the attached response to reviewers. We hope the way we addressed these concerns is acceptable to Reviewer 1 and the Editor. Thank you for the opportunity to reflect on these issues. 

Reviewer_1 Technical Comment #3 

A minor thing but I thought it was confusing that the samples a referred to by the country name. I would suggested referring to the „Ghanaian sample“ or „ sample collected in Ghana“ instead of just „Ghana“, for instance.

Authors’ Response:

Thank you for this feedback. We have made appropriate changes. 

We wish to thank the two reviewers for the constructive and positive feedback.

---

## [Decision Letter · Decision Letter 2]

9 Dec 2021

PONE-D-21-03687R2Effects of a Sport-Based Positive Youth Development Program on Youth Life skills and Entrepreneurial MindsetsPLOS ONE

Dear Dr. Malete,

Thank you for submitting your manuscript to PLOS ONE. After careful consideration, we feel that it has merit but does not fully meet PLOS ONE’s publication criteria as it currently stands. Therefore, we invite you to submit a revised version of the manuscript that addresses the points raised during the review process.

We look forward to receiving your revised manuscript.

Kind regards,

Ali B. Mahmoud, Ph.D.

Academic Editor

PLOS ONE

Journal Requirements:

Reviewers' comments:

Reviewer's Responses to Questions

**Comments to the Author**

1. If the authors have adequately addressed your comments raised in a previous round of review and you feel that this manuscript is now acceptable for publication, you may indicate that here to bypass the “Comments to the Author” section, enter your conflict of interest statement in the “Confidential to Editor” section, and submit your "Accept" recommendation.

Reviewer #1: (No Response)

2. Is the manuscript technically sound, and do the data support the conclusions?

Reviewer #1: Partly

3. Has the statistical analysis been performed appropriately and rigorously? 

Reviewer #1: Yes

4. Have the authors made all data underlying the findings in their manuscript fully available?

Reviewer #1: No

5. Is the manuscript presented in an intelligible fashion and written in standard English?

Reviewer #1: Yes

6. Review Comments to the Author

Reviewer #1: Thank you for addressing my comments and also providing a link to the data file, which is great. Unfortunately, this data file does not seem to contain all information relevant to perform the analyses reported in the sample. For instance, there is no information on the condition, the time of data collection (or how to tell which IDs belonged to the same person), and it only contains the LSSS but not the GET-variables. Please update this dataset so that an independent researcher would be able to understand the data set and replicate your analyses.

7. PLOS authors have the option to publish the peer review history of their article (what does this mean?). If published, this will include your full peer review and any attached files.

Reviewer #1: No

---

## [Author Response · Author response to Decision Letter 2]

9 Dec 2021

Reviewer #1: Thank you for addressing my comments and also providing a link to the data file, which is great. Unfortunately, this data file does not seem to contain all information relevant to perform the analyses reported in the sample. For instance, there is no information on the condition, the time of data collection (or how to tell which IDs belonged to the same person), and it only contains the LSSS but not the GET-variables. Please update this dataset so that an independent researcher would be able to understand the data set and replicate your analyses.

Authors Response: Accept my apology for uploading the link to wrong data set. I have uploaded the appropriate link and its: https://doi.org/10.6084/m9.figshare.17125589

---

## [Decision Letter · Decision Letter 3]

13 Dec 2021

Effects of a Sport-Based Positive Youth Development Program on Youth Life skills and Entrepreneurial Mindsets

PONE-D-21-03687R3

Dear Dr. Malete,

We’re pleased to inform you that your manuscript has been judged scientifically suitable for publication and will be formally accepted for publication once it meets all outstanding technical requirements.

Kind regards,

Ali B. Mahmoud, Ph.D.

Academic Editor

PLOS ONE

Additional Editor Comments (optional):

Reviewers' comments:

Reviewer's Responses to Questions

**Comments to the Author**

1. If the authors have adequately addressed your comments raised in a previous round of review and you feel that this manuscript is now acceptable for publication, you may indicate that here to bypass the “Comments to the Author” section, enter your conflict of interest statement in the “Confidential to Editor” section, and submit your "Accept" recommendation.

Reviewer #1: All comments have been addressed

2. Is the manuscript technically sound, and do the data support the conclusions?

Reviewer #1: Yes

3. Has the statistical analysis been performed appropriately and rigorously? 

Reviewer #1: Yes

4. Have the authors made all data underlying the findings in their manuscript fully available?

Reviewer #1: Yes

5. Is the manuscript presented in an intelligible fashion and written in standard English?

Reviewer #1: Yes

6. Review Comments to the Author

Reviewer #1: (No Response)

7. PLOS authors have the option to publish the peer review history of their article (what does this mean?). If published, this will include your full peer review and any attached files.

Reviewer #1: No

---

## [Editor Report · Acceptance letter]

15 Dec 2021

PONE-D-21-03687R3 

Effects of a Sport-Based Positive Youth Development Program on Youth Life skills and Entrepreneurial Mindsets 

Dear Dr. Malete:

I'm pleased to inform you that your manuscript has been deemed suitable for publication in PLOS ONE. Congratulations! Your manuscript is now with our production department. 

Kind regards, 

on behalf of

Dr. Ali B. Mahmoud 

Academic Editor

PLOS ONE